# TETRIS: TilE-matching the TRemendous Irregular Sparsity

**Yu Ji**[1,2,3]    **Ling Liang**[3]    **Lei Deng**[3]    **Youyang Zhang**[1]    **Youhui Zhang**[1,2]*    **Yuan Xie**[3]

{jiy15,zhang-yy15}@mails.tsinghua.edu.cn,zyh02@tsinghua.edu.cn
[1]Department of Computer Science and Technology, Tsinghua University
[2]Beijing Innovation Center for Future Chip
{lingliang,leideng,yuanxie}@ece.ucsb.edu
[3]Department of Electrical and Computer Engineering, University of California, Santa Barbara

## Abstract

Compressing neural networks by pruning weights with small magnitudes can significantly reduce the computation and storage cost. Although pruning makes the model smaller, it is difficult to get a practical speedup in modern computing platforms such as CPU and GPU due to the irregularity. Structural pruning has attracted a lot of research interest to make sparsity hardware-friendly. Increasing the sparsity granularity can lead to better hardware utilization, but it will compromise the sparsity for maintaining accuracy.

In this work, we propose a novel method, TETRIS, to achieve both better hardware utilization and higher sparsity. Just like a tile-matching game[2], we cluster the irregularly distributed weights with small value into structured groups by reordering the input/output dimension and structurally prune them. Results show that it can achieve comparable sparsity with the irregular element-wise pruning and demonstrate negligible accuracy loss. The experiments also show ideal speedup, which is proportional to the sparsity, on GPU platforms. Our proposed method provides a new solution toward algorithm and architecture co-optimization for accuracy-efficiency trade-off.

## 1 Introduction

Deep neural networks (DNNs) have achieved great success in a wide spectrum of applications, such as computer vision [1, 2, 3], speech recognition [4, 5], and language translation [6]. However, the huge memory overhead and intensive computation requirement limit the execution performance on cloud platforms and also impede the porting onto edge devices with constraints on resource and energy. Model compression by pruning can significantly reduce the storage and computation cost [7, 8, 9, 10, 11, 12]. They can achieve impressive compression rate by pruning weights with small magnitudes and then retraining the model to recover the accuracy.

Although pruning makes the model smaller, it requires specialized sparse BLAS library or customized hardware [13, 14, 15, 16, 17, 18] to accelerate its execution. On general computing platforms (e.g., CPU and GPU), the performance of the pruned sparse model may be even *worse* than the original dense one if the sparsity is not sufficiently high [19, 20]. This is because these general computing

platforms are usually optimized for continuous data access and computation. Sparse weight matrices lose the regular structure of dense matrices, it requires extra computation to decode the sparse format. In addition, it also creates irregular memory access, which is extremely slow in modern memory systems. Sparsity at this fine-grained granularity is not hardware-friendly.

Recent studies on structured sparsity [21, 22, 19, 23, 24, 20] have yielded better performance improvements. They usually group weight elements into small dense regions and prune them at the granularity of groups. Different kinds of grouping methods have been proposed to achieve better acceleration and higher sparsity. Many of them use coarse-grained groups such as channel-wise pruning and row/column-wise pruning [21, 22, 19, 23, 24]. These approaches usually shrink the size of some dimensions and the rest operations remain dense structure of smaller size. However, they usually suffer from less sparsity for maintaining accuracy. Some other studies [20] introduce very small groups with limited numbers of adjacent elements. They can achieve similar sparsity to the element-wise pruning but the performance increase is still far from the ideal expectation. Here the ideal performance means that the reduction in execution time is ideally proportional to the removed operations.

Simply increasing the sparse granularity can help improve hardware utilization but compromise the sparsity. Instead of carefully making tradeoffs between the two, we propose a reordering method to achieve both better hardware utilization and higher sparsity. The key idea is to cluster elements with small magnitude closer by reordering the input and output dimensions before pruning at a coarse granularity. It transforms irregularly distributed small elements into dense regions which can be removed for coarse-grained sparsity. Our method achieves comparable sparsity with irregular fine-grained pruning and maintains the model accuracy to a great extent. Meanwhile, we can achieve significant performance improvement due to the coarse-grained pruning granularity. The introduced overhead to reorder of input and output dimensions is negligible compared to the saved time.

It's worth noting that, our approach is orthogonal to other structured pruning methods. By reordering the input and output dimensions, we can always increase the sparsity of pruned networks generated by those pruning methods. In this paper, we take the block sparsity [25, 24] as a study case, but it is possible to extend to different structured sparsity pattern.

## 2  Related Work

Han *et al.* [8, 9] first proposed the deep compression and pruning approach, which reduced $> 90\%$ parameters on AlexNet and VGG16. However, it is very difficult to enjoy the benefits for speedup on GPU due to the irregular sparsity [19, 20]. Even implementing the deep compression through sparse matrix encoding on the specialized accelerators [13, 14], the indexing overhead is still costly. Moreover, the compressed model leads to irregular memory access pattern, which is extremely slow in modern memory systems. It is difficult to leverage the sparsity in NN for performance improvement.

The following studies tried efforts from different perspectives to produce structured sparsity. The studies on medium-grained sparsity (e.g. row/column level) [19, 26] presented a more regular pattern via L2-norm group-lasso optimization. However, this row or column sparsity is still not the favorite grain of general computing platforms. Coarser-grained sparsity (e.g. filter level) was obtained [10, 11, 12, 21, 22] through formulating and solving various optimization problems, such as filter scaling, group lasso, importance prediction, lower-rank forcing, etc. Nevertheless, the sparsity generated by these coarse-grained pruning methods is usually compromised because of the accuracy maintaining. Other studies [20] introduce very small groups with limited numbers of adjacent elements. They presented similar sparsity with the element-wise pruning but the performance improvement was still limited.

## 3  Sparsity Granularity

Pruning methods always try to find a boolean mask tensor $M$ to mark the pruned elements (1 for pruned elements; 0 for preserved elements) for a given weight tensor $W$ so that the number of elements in resulted weights $(W \Leftarrow (1 - M) \odot W)$ is minimized with certain structural constraints. Here $\odot$ represents element-wise multiplication. In this paper, we denote any pruning method for finding $M$ as a mapping $P(\cdot)$ that $M = P(W)$. Then, they retrain the model to recover accuracy, in which the pruned weights are constantly zero. The generation of $M$ is usually done by partitioning

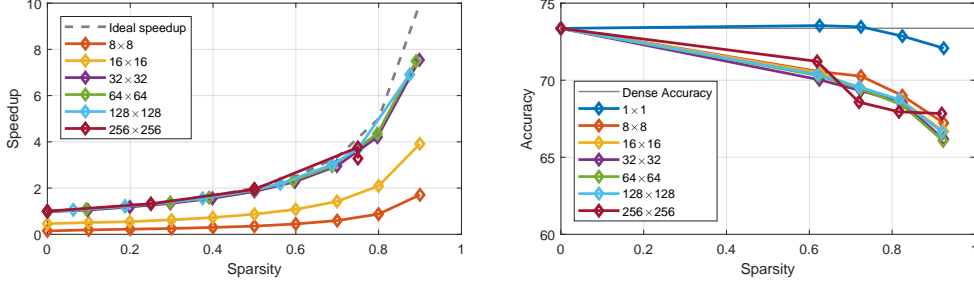

(a) Speedup vs. sparsity under different block size on conv4-2 layer from VGG16.

(b) Accuracy vs. sparsity under different block size on VGG16.

Figure 1: Tradeoffs between sparsity degree and hardware utilization with different pruning granularity.

the tensor into many dense regions and then selecting unimportant regions to prune. The size of the dense region is the granularity of sparsity.

The granularity of sparsity affects both the sparsity degree and the hardware utilization. We test the execution performance under different block size and sparsity. The *blocksparse* library [25], an open-source GPU kernel for block sparsity, is used for the evaluation on a Titan V GPU. Figure 1a shows the result of a $512 \times 512 \times 3 \times 3$ convolution on a $28 \times 28 \times 512$ input feature map, which is the most computation-intensive layer in VGG16 [1] for ImageNet [27] recognition. The baseline is the dense computation in Pytorch [28] with cuBlas as backend. The block sparsity is along the channel dimensions (512 here). Deep Compression [9] reported a sparsity of 73% for that layer, so we take this sparsity for detailed analysis. At this sparsity level, we can see that when the block size is less than 32, no practical performance improvement is gained. In contrast, when we set the block size to over 32, the speedup is approaching the ideal case. Therefore, the sparsity granularity, i.e. block size in this paper, should be sufficiently large for fully utilizing the hardware.

However, on the other side, when we increase the sparsity granularity too much, the accuracy will drop significantly. We also use VGG16 as an example to test the relationship between accuracy and sparsity under different block size. Since the dimensions of the first few layers are not large enough, we enforce at least two blocks for those small layers in our test. For example, for the first convolution layer with kernel dimension of $64 \times 3 \times 3 \times 3$, we use a block size of $32 \times 3 \times 3 \times 3$. As shown in Figure 1b, even if we only set a small block size $8 \times 8$, the accuracy still drop significantly.

Although we use block sparsity as an example, the trade-off between accuracy and hardware utilization (determined by sparse pattern and sparsity) is prevalent. The unimportant elements are usually irregularly distributed while grouping them toward regular pattern will greatly restrict the flexibility of selecting those unimportant elements and then compromise the sparsity for maintaining accuracy.

## 4 Reordering Irregular Sparsity

Instead of struggling to achieve a good trade-off among the sparsity, hardware utilization, and the accuracy, and then selecting a proper grouping granularity, we propose an orthogonal approach that clusters unimportant elements together to form regular structures. It enables the structured pruning algorithms to achieve both higher sparsity and lower accuracy loss.

Let $W \in \mathbb{R}^{m \times n}$ and $B \in \mathbb{R}^n$ denote the weight matrix and bias vector of a fully-connected (FC) layer, the number of input and output neurons is $m$ and $n$, respectively. If we feed a batch of inputs $X \in \mathbb{R}^{N \times m}$ with $N$ samples, we can get $Y = \sigma(WX + B)$, where $\sigma$ is an element-wise activation operation. Now we introduce two permutations $\alpha$ and $\beta$ for the two dimensions of $W$:

$$\alpha = \begin{pmatrix} 1 & 2 & \dots & m \\ a_1 & a_2 & \dots & a_m \end{pmatrix} \quad \beta = \begin{pmatrix} 1 & 2 & \dots & n \\ b_1 & b_2 & \dots & b_n \end{pmatrix}. \tag{1}$$

Then, the layer computation can be governed by

$$Y[I; \beta] = \sigma(W[\alpha; \beta]X[I; \alpha] + B[\beta]) \tag{2}$$

where $W[\alpha; \beta]$ denotes the matrix in which the rows and columns are reordered according to permutations $\alpha$ and $\beta$, and $I$ is the unit permutation. After reordering, we can first apply $\alpha$ on

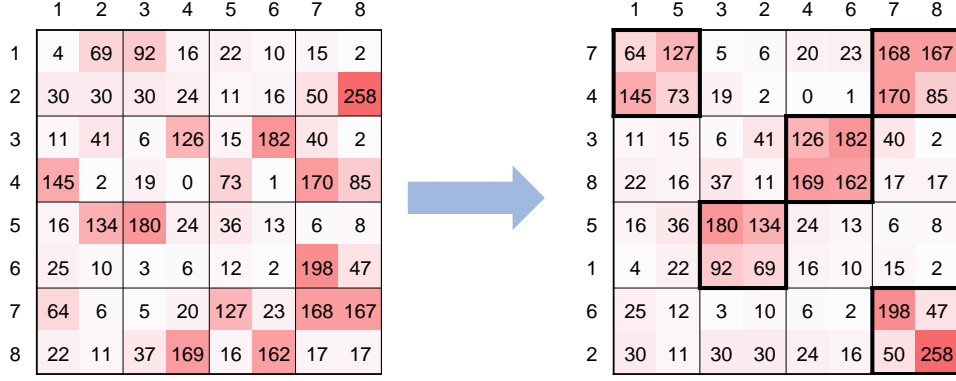

Figure 2: Reordering to cluster elements with similar magnitude for structured pruning.

$X$, and feed it into a new fully-connected layer with weight matrix $W[\alpha; \beta]$ and bias $B[\beta]$ to get $Y[\beta]$. Then we only need to apply the reverse permutation $\beta^{-1}$ to get the original result $Y$. The overhead introduced during runtime is only the permutation on input and output. But it provides us the flexibility to choose a good $\alpha$ and $\beta$ to cluster irregularly distributed small elements together so that we can gain more sparsity for better hardware utilization with less accuracy loss.

For convolution, we can also reorder the weights as follows

$$Y[I; \beta; I; I] = \sigma(W[\beta; \alpha; I; I] \otimes X[I; \alpha; I; I] + B[\beta]) \tag{3}$$

where $W$ is the weight kernel, $X$ is the input feature maps, $Y$ is the output feature maps, and $B$ is the bias vector. $\alpha$ and $\beta$ are permutations on the input-channel and output-channel dimensions, respectively.

As shown in Figure 2, we can leverage the flexibility of permutation and properly reorder the indices for each dimension to cluster elements according to the value magnitude, which enables a better sparsity for structured pruning algorithms.

## 4.1 Reordering Algorithm

For generality, we assume that the weight tensor has $d$ dimensions. A pruning method $P$ that generates a mask $M = P(W)$ usually tries to find a good mask so that the norm of the pruned value is minimized.

$$P(W) = \arg\min_M \|M \odot W\| \quad s.t. \quad M \text{ satisfies structural constraints} \tag{4}$$

The norm used here depends on the pruning algorithm.

In this paper, we introduce a new dimension, permutations of the weight tensor, to further minimize the objective. We have the opportunity to choose the best permutations $\alpha_1, \ldots, \alpha_d$ to minimize the pruned values, i.e.

$$\min_{M, \alpha_i, i \in \Omega} \|M \odot W[\alpha_1; \ldots; \alpha_d]\| \quad s.t. \quad M \text{ satisfies structural constraints} \tag{5}$$

The dimensions that can be reordered are denoted as set $\Omega$, and others are set to the unit permutations.

Analytically solving the minimization problem is difficult. The target is to generate permutations that cluster unimportant elements closer into structured dense regions. This is similar to a k-means problem that can be iteratively solved by the expectation maximization (EM) algorithm.

- E-Step: Fix the permutation $\alpha = \alpha^{(t)}$ and generate a mask by using the given pruning method $P$, i.e.

$$
\begin{aligned}
M^{(t+1)} &= \arg\min_M \left\| M \odot W[\alpha_1^{(t)}; \ldots; \alpha_d^{(t)}] \right\| \quad s.t. \quad M \text{ satisfies structural constraints} \\
&= P(W[\alpha_1^{(t)}; \ldots; \alpha_d^{(t)}])
\end{aligned}
\tag{6}
$$

- M-Step: Fix the mask $M^{(t+1)}$ and find optimized permutations $\alpha_i^{(t+1)}$ such that the masked values are minimized, i.e.

$$\alpha_1^{(t+1)}, \ldots, \alpha_d^{(t+1)} = \arg \min_{\alpha_i, i \in \Omega} \left\| M^{(t+1)} \odot W[\alpha_1; \ldots; \alpha_d] \right\| \tag{7}$$

.

We can use unit permutations as the initial configuration for all dimensions and run above EM algorithm iteratively until it converges. However, the M-Step is still an optimization problem. Since the permutations of different dimensions are highly coupled, it is difficult to generate optimized permutations for all dimensions. We use an alternating minimization (AM) algorithm to optimize different dimensions separately and iteratively. Each time, we fix other dimensions and only optimize one dimension $D$, which can be described as

$$\arg \min_{\alpha_D} \| M \odot W[\alpha_1; \ldots; \alpha_d] \| . \tag{8}$$

The possible permutations still have a large search space. We start from the unit permutation, and greedily swap two indices that can mostly decrease equation (8) each time until the convergence. Finding the index pair highly depends on the exact form of the norm in the pruning algorithm $P$. For example, if L1 norm is employed, then we use the absolute value of $W$ as its importance matrix. For L2 norm, the importance matrix is the square of $W$. Without loss of generality, we use L1 norm as an example. We first contract the importance matrix of $W$ with $M$ along all dimensions except the $D$-th dimension, i.e.

$$S_{ij} = \sum_{k_1, \ldots, k_{D-1}, k_{D+1}, \ldots, k_d} |W_{k_1, \ldots, k_{D-1}, i, k_{D+1}, \ldots, k_d}| M_{k_1, \ldots, k_{D-1}, j, k_{D+1}, \ldots, k_d} \tag{9}$$

The result $S$ is a square matrix whose size is the same as the current dimension $D$. The element $S_{ij}$ represents the total value of masked elements wherein we mask the $i$-th slice in $W$ with the $j$-th slice in $M$. Thus, the decrease of the objective function that we can gain from swapping the $i$-th slice and the $j$-th slice is $G_{ij}$ as shown in Equation (10), which can also be written in the matrix format (11), where $L$ is the diagonal vector of $S$

$$G_{ij} = S_{ii} + S_{jj} - S_{ij} - S_{ji} \tag{10}$$

$$G = L + L^T - S - S^T \tag{11}$$

Then, we only need to find the maximum elements $G_{ij}$ in $G$ and swap the $i$-th slice and $j$-th slice in $D$ dimension of $W$. The algorithm is described as in Algorithm 1.

---
**Algorithm 1** Reordering algorithm
---
**Input:** Pruning Method $P$, $d$-order weight tensor $W$, reordering dimension set $\Omega$
**Output:** Permutations $\alpha_i$
    Initialize all $\alpha_i = I$
    **repeat**
        $M = P(W)$
        **for** $D \in \Omega$ **do**
            **repeat**
                Compute tensor contraction $S = |W|M$ over dimensions $\{1, \ldots, D-1, D+1, \ldots, d\}$.
                $L = diagonal(S)$
                $G = L + L^T - S - S^T$
                $i, j = \arg \max(G)$
                Swap $i$-th and $j$-th slices in $D$-th dimension of $W$
                Swap $i$-th and $j$-th indices of $\alpha_D$
            **until** $\max(G) \leq \epsilon$ where $\epsilon$ is a small enough positive number.
        **end for**
    **until** Convergence
---

## 4.2 Pruning Overhead Optimization

The algorithm includes three nested iterations, in which the inner loop takes more iterations to converge than the other two. The inner loop contains a tensor contraction, which is computationally intensive. Although we only need to run the pruning algorithm once before fine-tuning, the overhead is still too large. Fortunately, there exists large computational redundancy so that we can reuse many intermediate results from the previous iteration. At each iteration during the inner loop, the only update is to swap the $i$-th and $j$-th slices in the $D$-th dimension of $W$. Thus, for $S$, we only need to swap the $i$-th and $j$-th rows without recomputing the tensor contraction. Consequently, we can move the computational intensive tensor contraction to the outer loop.

The rest of the computations in the inner loop are all element-wise operations or max operations, whose computational complexity is proportional to the size of the matrix $S$. However, for some large layers, the size of the corresponding $S$ is $n^2$, which is still time-consuming to perform element-wise operations or max operations over $S$. For example, the first fully-connected layer of VGG16 [1] has an input of size 25088, and our algorithm consumes more than 2 hours. We can further optimize these $O(n^2)$ operations as follows: (i) Since the update on $S$ is only to swap two rows $i$ and $j$, $G$ only needs to recompute the $i$-th and $j$-th rows and columns, which is $O(n)$ complexity. (ii) To optimize the computation of finding the maximum value in $G$, we maintain two vectors of size $n$ to record the maximum value in each row of $G$ and their indices. Each time when we recompute the $i$-th and $j$-th rows and columns, we need first recompute the maximum value of the $i$-th and $j$-th rows, which is also $O(n)$ complexity. For the rest rows, if the original maximum value is not in the $i$-th or $j$-th columns, we only need to compare the new value in the two columns with the original maximum value of each row, which is also $O(n)$ complexity. Otherwise, we recompute the maximum value of those rows; the complexity is $O(nr)$, where $r$ is the number of elements in the $i$-th or $j$-th column that holds the maximum values of their rows. In practical, $r$ is usually far less than $n$. With these optimizations, we can prune the entire large VGG16 model in less than 1 minute on a TiTan V GPU, which can be ignored compared to the fine-tuning time.

## 4.3 Runtime Overhead Optimization

The introduced overhead in the inference phase is that we have to reorder the input and output of all layers according to the generated permutations. It is a pure data-movement operation from a dense tensor to another, which can fully utilize the bandwidth of GPU. It only takes about $4\%$ of the computation time of the normal layers. Considering the benefits it brought, this small overhead is also negligible.

In addition, since the layers between two adjacent weighted layers are usually activation function or pooling operation along kernel dimensions (not the permuted dimensions along the channels), we can merge the output permutations of the previous layer with the input permutations of the next one. Thus, on average, each layer only requires one reordering data movement. With the optimized runtime overhead, we are able to speed up the computation of the layers close to the ideal case.

## 5 Experiments

Our reordering algorithm is general for structural pruning methods. Note that, our reordering algorithm is orthogonal to existing structural pruning algorithm. Thus, we use one typical structural pruning algorithm, block sparsity, as a case study to show how our algorithm can improve its sparsity and granularity.

Block sparsity is to first partition the weight tensor into small blocks with a grid and then prune the weight tensor at the granularity of blocks. In our experiment, we use the L1 norm of each block to measure the importance of the block and prune the blocks with smaller importance value. The block sparsity without reordering is our baseline algorithm.

We implement our reordering and pruning method in Pytorch [28]. The reordering algorithm will generate a mask and permutations for each weighted layer. We reorder the mask according to the reverse permutations to generate a permuted mask and use it to filter the elements of the original weights so that we do not need to reorder the inputs and outputs for retraining. We test our method on three networks of different scales: LeNet on MNIST, VGG14 on CIFAR-10, and VGG16 on ImageNet. The first two models are trained from scratch to get the baseline accuracy, and the last

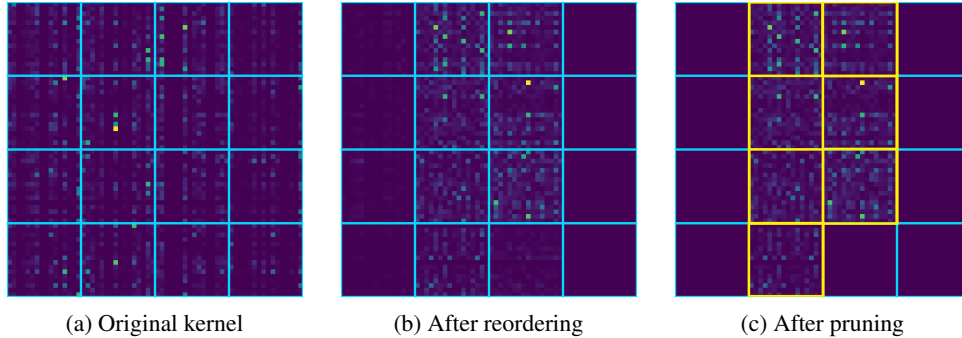

| (a) Original kernel | (b) After reordering | (c) After pruning |

Figure 3: Reordering and pruning.

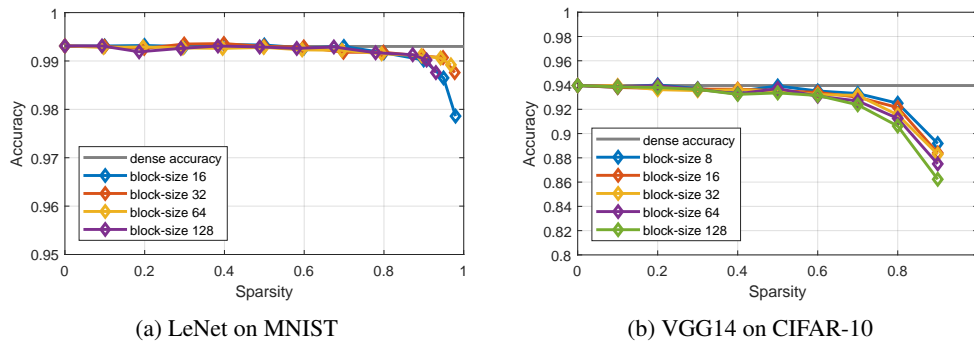

(a) LeNet on MNIST

(b) VGG14 on CIFAR-10

Figure 4: Sparsity vs. Accuracy under different block sizes for LeNet on MNIST and VGG14 on CIFAR-10.

model is obtained from torchvision [29]. For retraining the pruned VGG16, we use a learning rate of 0.001 and retrain 20 epochs. In all of our tests, if the block size is larger than the channel dimension of the layer, we will reduce the block size for that layer to ensure that each layer at least has two blocks.

Figure 3 visualizes the process of reordering and pruning the second convolutional layer in VGG16. We sum up the kernel dimensions to visualize it as a 2D image. Figure 3a is the original weights. After reordering, the elements are distributed as in Figure 3b. Then, we can prune the blocks with small values to get the final weights in Figure 3c.

## 5.1 Models on MNIST and CIFAR-10

We first do experiments on one small model, LeNet on MNIST, and one medium model, VGG14 on CIFAR-10. The accuracies for the original LeNet and VGG14 are $99.3\%$ and $93.96\%$, respectively. Figure 4 shows the relationship between sparsity and accuracy under different block sizes. From the figure, we can see that block size has little impact benefit because of our reordering method. We can achieve about $10\times$ and $3\times$ speedup on the two models, respectively.

Note that, for simplicity, we set the same pruning rate for all layers, which is not the optimal configuration. The accuracy can be improved by setting a lower pruning rate for some sensitive layers (e.g. the first and the last layer). However, searching the hyper-parameter space to improve the sparsity-accuracy curve is orthogonal to our approach. Our contribution is to make the curves of larger block sizes closer to that of smaller block sizes. In this way, we enable great performance improvement with only little accuracy degradation.

## 5.2 VGG16 on ImageNet

We also test our approach on a large-scale model, VGG16 on ImageNet dataset. For simplicity, we use the pruning rate from Deep compression [9] as the basic configuration. They can prune $92\%$ of

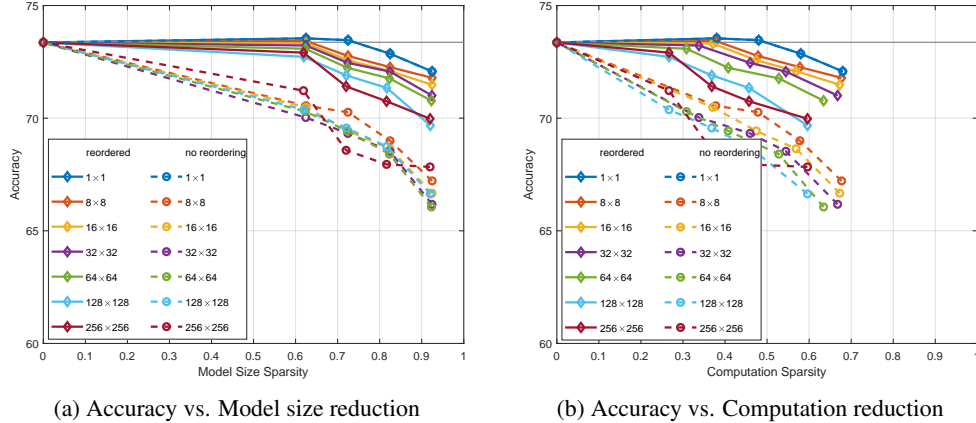

(a) Accuracy vs. Model size reduction          (b) Accuracy vs. Computation reduction

Figure 5: Top-1 accuracy vs. Sparsity under different block sizes for VGG16 on ImageNet. Without our reordering method, the accuracy for all case will drop $2.3\% \sim 6.0\%$ compared to the $1 \times 1$ case. With reordering, the accuracy drop will decrease to $0.1\% \sim 2.4\%$.

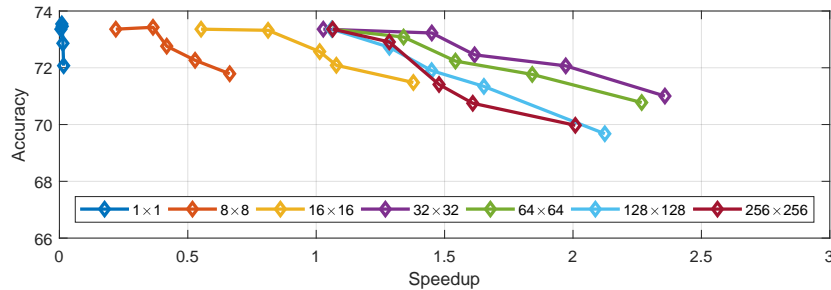

Figure 6: Top-1 accuracy vs. Speedup under different sparsity and block sizes for VGG16 on ImageNet. For each block size, we test the computation sparsity from $0\%$ to $68\%$. For the best block size configuration, $32 \times 32$, the accuracy will decrease from $73.36\%$ to $71.004\%$. Compared to the $1 \times 1$ case that decrease to $72.074\%$, the accuracy drop is small. But the speedup will increase to $2.35\times$ compared to the dense baseline while the performance of $1 \times 1$ case is much worse than the baseline.

the model size and $68\%$ of the operations of VGG16. In addition, we test the configurations in which the pruning rates of all layers gradually decreases by $10\%$, $20\%$, and $30\%$.

Figure 5 shows the relationship between sparsity and accuracy. Figure 5a is the sparsity in terms of model size and Figure 5b is the sparsity in terms of computation. From the perspective of acceleration, the sparsity of computation is more relevant. Compared to the baseline without reordering that the accuracy significantly dropped, our approach can make the curve of block-wise pruning close to the element-wise case ($1 \times 1$ case).

## 5.3  Speedup vs. granularity

The granularity of sparsity affects the speedup from two aspects: coarser granularity leads to better hardware utilization, which pushes the practical speedup closer to the ideal case; but it would impair the pruning rate. For example, in Deep Compression [9], the pruning rate for the first layer in VGG16 is $42\%$. However, when the block size increases to 32, the whole layer only consists of two blocks. We can only prune nothing or $50\%$ of the weights. For the similar layers, we have to decrease the block size that leads to smaller sparsity and less speedup.

As shown in Figure 6, we plot the relationship between accuracy and speedup under different block sizes for VGG16. As the block size increases the speedup first increases until block size reaches 32, then it begins to decrease. The critical point is caused by two factors. Typically, for most convolutional layers in VGG16, we can almost make full use of hardware when the block size is larger than 32. If we keep increasing the block size, the sparsity may decrease because of the limited

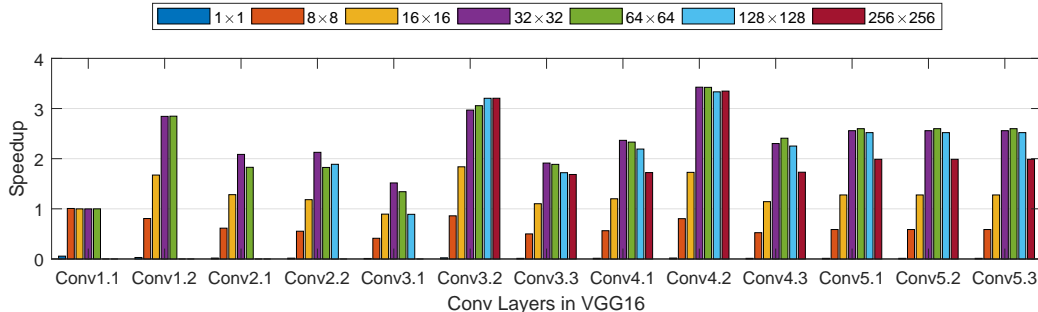

Figure 7: Speedup of convolutional layers in VGG16 with different block sizes based on Blocksparse library [25]. The baseline is the dense implementation in pytorch based on cuBlas.

pruning rate for maintaining accuracy. Since the speedup is inversely proportional to the amount of remained computation, a small change on the pruning rate may lead to a significant decrease in speedup.

Different layers have different critical points. Figure 7 shows the speedup of different convolutional layers in VGG16 under different block sizes; and the pruning rate configuration is set to the same as that of Deep Compression. For layers that have fewer channels, they achieve the best speedup when the block size is set to 32. For layers with more channels, the critical point is 64.

## 6    Conclusion

The coarse-grained sparsity is usually beneficial to achieve higher speedup on parallel hardware, but it usually achieves less sparsity or accuracy compared to the fine-grained sparsity. In this paper, we present a method to reorder irregular fine-grained sparsity to structured coarse-grained sparsity to bridge the gap between the large sparsity we can gain from models and the poor practical speedup. It can also help the fine-grained pruning methods to achieve the ideal execution acceleration.

## Acknowledgement

This research was collaborative work of Tsinghua University and University of California, Santa Barbara. Thanks for the support from Beijing Innovation Center for Future Chip, Science and Technology Innovation Special Zone project, and the National Science Foundations (NSF) under grant numbers 1725447 and 1730309. We also thank OpenAI for their open-source library, blocksparse.

## Footnotes

[2] A tile-matching game is a type of game where the player manipulates tiles in order to make them disappear according to a matching criterion. **Tetris** is one of the most famous tile-matching games. Our approach is doing the same thing that clusters the unimportant items and structurally prunes them.

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
