[Reviews · NeurIPS 2018]

Reviewer 1



This paper deals with sparsifying the weights of a neural net to reduce memory requirements and speed up inference. Simply pruning small weights yields unstructured sparsity, which is hard to exploit with standard libraries and hardware. This paper imposes block sparsity, where each weight tensor is divided into fixed blocks (of size 32 x 32, for example) and non-zero weights are specified in only a fraction of the blocks. The paper's innovation is an iterative algorithm for reordering the rows and columns of a tensor to group together the large weights, reducing the accuracy loss from block pruning. Experiments on the VGG16 network for ImageNet show that this method achieves better speed-accuracy trade-offs than either unstructured weight pruning or block pruning without reordering. Update: I appreciate the authors' responses. I've added some more comments in the body of my review, marked with the word "Update". Summary of evaluation: The paper shows solid empirical results for pruning network weights in a way that doesn't lose much accuracy and actually speeds up inference, rather than slowing it down due to indexing overhead. The proposed reordering algorithm could use more analysis and could be better related to previous work. On balance, though, I think the paper is worth publishing. Originality: The paper builds on a preprint by Narang et al. (2017) that proposes block-level pruning for neural nets without row/column reordering. There also seems to be a large literature in the computational linear algebra community on reordering rows and columns to make better use of sparsity. Examples of that work include: P. R. Amestoy, T. A. Davis, I. S. Duff (1996) "An approximate minimum degree ordering algorithm". SIAM J. Matrix Analysis & Applic. 17(4):886-905. A. Pinar, M. T. Heath (1999) "Improving performance of sparse matrix-vector multiplication". Proc. ACM/IEEE Conf. on Supercomputing. Some of that literature is summarized with diagrams at https://www.mathworks.com/help/matlab/examples/sparse-matrices.html. I didn't find any prior work tackling the exact problem being addressed here, but it would be helpful to consider how insights from that literature might be useful for the current problem. Update: The authors' response mentions that their algorithm was inspired by a 2010 paper (on a different problem) by Goreinov et al. I couldn't find a reference to that paper in the current manuscript; it would be good to include it. Clarity: For the most part, the paper is easy to follow. A few points could be made clearer. First, it's hard to tell exactly what base pruning method was used in the experiments. Based on a comment at the end of the introduction (p. 2, line 54), I assume it's the method of Narang et al. (2017). It would be helpful to describe that method explicitly. It would also be helpful to say exactly what is meant by a block-sparse matrix here: a key point is that a regular grid of blocks is fixed ahead of time (if I understand correctly from the blocksparse library documentation), in contrast to, say, a general block-diagonal matrix, where the blocks can have arbitrary sizes and alignments. Update: I appreciate the discussion of these points in the response. In the experiment section, Fig. 6 makes it easy to see the comparison to the unstructured-pruning baseline, but it's harder to compare to the baseline of block pruning without reordering -- I needed to cross-reference with the accuracy numbers in Fig. 5(b). It would be good to include the no-reordering curves in Fig. 6 as well, at least for 32x32 blocks. Quality: The empirical results are good, showing a dramatic speed-up over the unstructured sparse baseline and a significant accuracy improvement compared to the baseline without reordering. The results are clearest for VGG16 on ImageNet. The paper also gives results for MNIST and CIFAR-10, suggesting that the improvement is not specific to one network. It would be even more persuasive to show wall time speed-up numbers for those networks as well (in addition to the sparsity-accuracy curves). Update: the authors' response reports 10x and 3x speed-ups for MNIST and CIFAR; it will be good to include those numbers in the paper. The quality of the algorithm section is mixed. On one hand, the row/column reordering algorithm makes intuitive sense. It's interesting to see the optimizations in Section 4.2 (p. 5-6) that reduced the running time of reordering from over 2 hours to under 1 minute for the VGG16 model. On the other hand, there's no discussion of consistency between the objective being optimized by the base pruning method P and the one being optimized by the reordering steps in the algorithm. It looks like the pruning method of Narang et al. (2017) minimizes the maximum magnitude among the pruned weights. Update: The authors' response includes the crucial fact that they switched to an L1 version of the Narang et al. pruning algorithm, minimizing the sum of absolute values of the pruned weights. This definitely needs to be stated in the paper. It's less clear what the reordering steps are optimizing, because Eq. 4 (p. 4) uses norm notation on the matrix of pruned weights without specifying what norm is intended. Based on Eq. 7, which sums the absolute values of entries, it seems to be the l1 norm. (Update: the authors' response confirms this.) So the base pruning method is optimizing a max over weight magnitudes, while the reordering steps are optimizing a sum. Could the algorithm end up oscillating, with the two steps undoing each others' work? Update: The response clarifies that, in fact, both parts of the algorithm are optimizing a sum, so there's no risk of oscillation. The paper should mention this explicitly. Conversely, if the base pruning objective is consistent with the reordering objective, is it really necessary to re-run pruning after the reordering steps? The reordering steps are taking the block choices from the previous pruning step and trying to make them as good as possible, so it's not clear whether another choice of blocks could end up even better. Update: The authors' response says that re-pruning is indeed necessary. It's obvious that an arbitrary reordering of rows and columns could necessitate re-pruning, but it's still not obvious to me that this can happen when both the reordering and the pruning are optimizing the L1 objective. Would it be possible to provide a small example where this happens, maybe in the Supplement? Significance: There is a lot of interest in running neural nets on devices with limited memory and computation resources. This work seems to make a useful contribution on that front. It is complementary to other methods such as discretization of weights; I'm not sure if there are other sparsification techniques intended for standard hardware that should be included as additional baselines for comparison.

Reviewer 2



This paper suggests a clustering method as a prelude to compression of neural network representations by weight pruning. This method strikes a middle between the coarse and granular pruning methods based on sparsity and allows for a better affinity with modern hardware and linear algebra libraries by reducing branching. The exposition of the paper is clear and the related work is explained and cited, though I'm not particularly familiar with this sub-domain. The presentation of the optimization problem for the input and output permutations that are sought in the reordering algorithm, as a k-means optimization is conveyed clearly, and the derivation of the successive steps of the greedy algorithm is straightforward. This reviewer would have loved to survey source code to check the details of the algorithm in action. The accounting of the cost of reordering input and output of layers is correct in accounting a modest cost overall, compared to the time necessary in fine-tuning the model. However, in a distributed setting using model parallelism, this re-sharding of the weights across a set of workers may prove more costly. The experimental part of the paper accounts for the trade-off between model size and accuracy, and that between speedup and granularity at different block sizes, and offer insightful learnings on inflexion points in the speedup gains as well as hints at optimizations for layer-wise choices of pruning rates. Overall a very good paper. Minor points: - the paper lacks the use of articles in various places English grammar would demand them, and could take a round of editing. - the end of section 1 and that of section 2 would gain by being better merged. Section 1 l.38-41 and Section 2 l.70-73 repeat each other.

Reviewer 3



The paper uses structured sparsity methods to overcome the trivial practical speedup of DNN inference by connection pruning because of the irregular sparse data pattern. Existing structured sparsity methods can learn regular sparse pattern and achieve higher speed, but they generally have lower sparsity because they introduce stronger optimization constraints. The paper reorders/clusters redundant/small weights into neighbor regions so that useless weights can be effectively removed simutaniously, and thus improve the structured sparsity and achieve higher speedup. A greedy method is proposed to search the permutation for reordering. Strengths: (1) It makes sence to reorder/cluster small weights and remove then together. In this way, a higher structured sparsity and thus higher speedup should be achieved. The idea is interesting. (2) It may further advance DNN inference acceleration for structured sparsity methods. However, the quality would have been better delivered if the following weaknesses were solved: (1) brute force (gready) search is simple and may work, but it's more interesting to compare it with existing clustering methods (e.g. biclustering) (2) reordering trained DNNs is straightforward, but intergrating it into the training/pruning process may be more interesting and more effective (3) the paper should compare with existing structured sparsity methods (like group Lasso) Minors and clarity: (1) "For generality, we assume that the weight tensor has d dimensions." Please clarify "d dimensions". For a tensor with R^{MxNxK}, sometimes people call it 3D but sometimes MxNxK dimensions. (2) clarify and define M = P(W). (3) "From the figure we can see that, block size has little impact benefit from our reordering method." however, there is certain correlation between block size, sparsity and accuracy. The correlation is more obvious in Figure 5.